# Clinical implications of the log linear association between LDL-C lowering and cardiovascular risk reduction: Greatest benefits when LDL-C >100 mg/dl

**Jennifer G. Robinson**[1]*, **Manju Bengaluru Jayanna**[2], **C. Noel Bairey Merz**[3], **Neil J. Stone**[4]

1 Division of Cardiology, Department of Epidemiology and Internal Medicine, University of Iowa, Iowa City, IA, United States of America, 2 Division of Cardiovascular Disease, Department of Medicine, Lenkenau Medical Center & Lankenau Institute for Medical Research, Wynnewood, PA, United States of America, 3 Barbara Streisand Women's Heart Center, Cedars-Sinai Smidt Heart Institute, Los Angeles, CA, United States of America, 4 Bluhm Cardiovascular Institute, Feinberg School of Medicine, Northwestern University, Chicago, IL, United States of America

* jennifer-g-robinson@uiowa.edu

**Data Availability Statement:** Data are included in the Supporting Information files - S1 Files 1 & 2

## Abstract

### Background

The log linear association between on-treatment LDL-C levels and ASCVD events is amplified in higher risk patient subgroups of statin versus placebo trials.

### Objectives

Update previous systematic review to evaluate how the log linear association influences the magnitude of cardiovascular risk reduction from intensifying LDL-C lowering therapy.

### Methods

MEDLINE/PubMED, Clinical trials.gov, and author files were searched from 1/1/2005 through 10/30/2019 for subgroup analyses of cardiovascular outcomes trials of moderate versus high intensity statin, ezetimibe, and PCSK9 mAbs with an ASCVD endpoint (nonfatal myocardial infarction or stroke, cardiovascular death). Annualized ASCVD event rates were used to extrapolate 5-year ASCVD risk for each treatment group reported in subgroup analyses, which were grouped into *a priori* risk groups according to annualized placebo/control rates of ≥4%, 3–3.9%, or <3% ASCVD risk. Data were pooled using a random-effects model. Weighted least-squares regression was used to fit linear and log-linear models.

### Results

Systematic review identified 96 treatment subgroups from 2 trials of moderate versus high intensity statin, 2 trials of a PCSK9 mAb versus placebo, and 1 trial of ezetimibe versus placebo. A log linear association between on-treatment LDL-C and ASCVD risk represents the association between on-treatment LDL-C levels and ASCVD event rates, especially in

**Funding:** The author(s) received no specific funding for this work.

**Competing interests:** Jennifer G Robinson MD MPH has received research grants to Institution: Acasti, Amarin, Amgen, Astra-Zeneca, Esperion, The Medicines Company, Merck, Novartis, Novo-Nordisk, Regeneron, Sanofi, Takeda and served as a consultant for Getz Pharma,The Medicines Company, Novartis, and Pfizer. Manju Bengularu Jayanna MBBS None. C Noel Bairey Merz MD has received research grants to Institution: Caladrius, Gilead, Sanofi ACT, and served as a consultant for iRhythm, Sanofi Vascular, and Abbott Diagnostics. Neil J Stone MD None. This does not alter our adherence to PLOS ONE policies on sharing data and materials.

higher risk subgroups. Greater relative and absolute cardiovascular risk reductions from LDL-C lowering were observed when baseline LDL-C was >100 mg/dl and in extremely high risk ASCVD patient groups.

## Conclusions

Greater cardiovascular and mortality risk reduction benefits from intensifying LDL-C lowering therapy may be expected in those with LDL-C $\geq$ 100 mg/dl, or in extremely high risk patient groups. When baseline LDL-C <100 mg/dl, the log linear association between LDL-C and event rates suggests that treatment options other than further LDL-C lowering should also be considered for optimal risk reduction.

## Introduction

Current guidelines recommend intensifying low density lipoprotein cholesterol (LDL-C) lowering therapy when LDL-C remains above certain thresholds, or to achieve LDL-C goals [1, 2]. However, the benefits of more aggressive LDL-C lowering have long been debated [3]. Epidemiologic data and the Cholesterol Treatment Trialists individual meta-analysis of the statin trials supports a log linear rather than linear association between total cholesterol or LDL-C level and observed cardiovascular event rates [4]. Confusion has arisen regarding this relationship because the log linear association appears linear when plotted on a log scale, or when plotted as ratios or percent difference [4, 5]. In addition, amplification of the log linear association between LDL-C levels and cardiovascular event rate occurs in higher risk patient subgroups in the statin versus placebo randomized trials and appears to be linear in lower risk patient group (**Fig 1A**) [6].

Evidence of a log linear association between LDL-C level and cardiovascular events for ezetimibe and PCSK9 mAb trials comes from a meta-analysis of statin, ezetimibe, and proprotein convertase subtilisin/kexin type 9 (PCSK9) inhibiting monoclonal antibody (mAb) cardiovascular outcomes trials [7]. In this meta-analysis, each 40 mg/dl higher LDL-C level at baseline was associated with an additional 9% reduction in all-cause mortality, 14% reduction in cardiovascular mortality, and 10% reduction in major adverse cardiovascular events. In addition, amplification of the log linear association in higher risk patient groups was found in a recent pooled analysis of Phase 3a efficacy trials of alirocumab, a PCSK9 mAb. An amplification of the log linear relationship between baseline LDL-C level and cardiovascular event rates was observed in very high risk patient groups with diabetes, chronic kidney disease or polyvascular disease (**S1 Fig**) [8].

Because a log linear association between LDL-C level and cardiovascular event rate will likely influence the magnitude of benefit from more aggressive LDL-C lowering therapy from ezetimibe or PCSK9 mAbs, we undertook this systematic review to update a previous analysis of subgroups from trials of statin versus placebo [6]. The results of this study were also considered in context of other drug therapies that have been shown to reduce atherosclerotic cardiovascular disease (ASCVD) risk in an effort to inform prioritization of drug therapies for high risk patients [9, 10].

## Methods

Methods are those used in a previous systematic review of statins, ezetimibe, and PCSK9 mAb trials with ASCVD outcomes by JGR and colleagues [11] (**S1 File**); inclusion dates were 1/1/

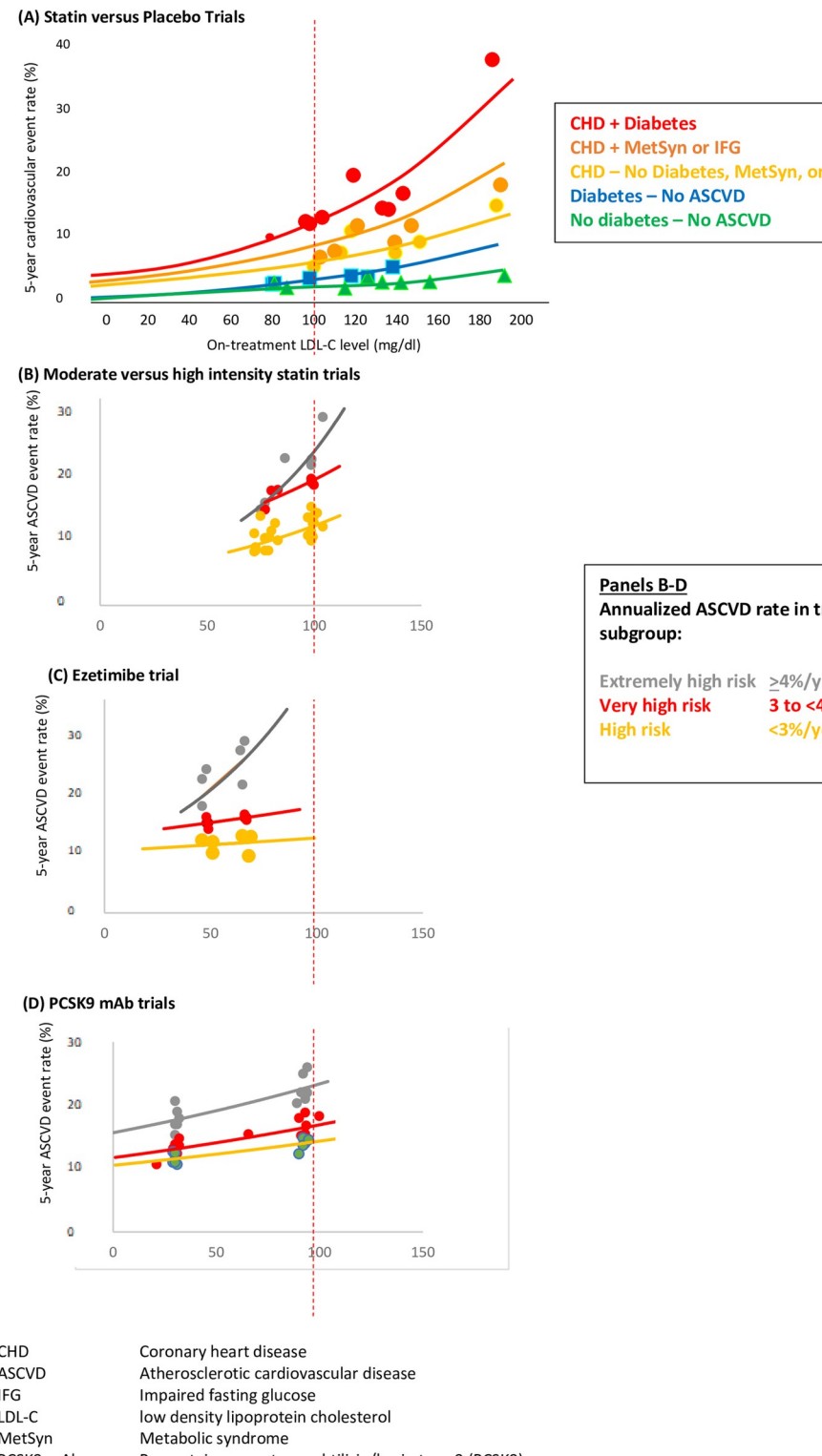

**Fig 1.** Achieved low density lipoprotein cholesterol (LDL-C) level and risk of cardiovascular disease (CVD) or atherosclerotic CVD (ASCVD) events by (A) subgroups from placebo-controlled statin trials of approximately a mean 5 years duration in the presence of coronary heart disease (CHD), metabolic syndrome (MS), impaired glucose tolerance (IGT) or diabetes (Adapted from Robinson JG, Stone NJ. Am J Cardiol 2006; 98:1405–1408); and subgroups categorized according to extrapolated 10-year ASCVD event rates from (B) moderate versus high intensity statin trials of a mean

3–5 years duration extrapolated to 5 years, and (C) PCSK9 inhibiting monoclonal antibody trials with a mean 2.2–2.8 years duration extrapolated to 5 years; Line represents log linear regression line for relationship between on-treatment LDL-C level and cardiovascular event rate weighted by group size.

2005 through 10/30/2019. The PRISMA checklist is provided in the Supplement (**S1 Table**). To facilitate comparison between trials, the composite ASCVD endpoint defined by the trial was used: fatal and nonfatal myocardial infarction (MI), fatal and nonfatal stroke and cardiovascular death or coronary heart disease (CHD) death if cardiovascular death not available). Cardiovascular death includes deaths from CHD and stroke. Annualized ASCVD event rate was calculated as the observed rate for the duration of the trial divided by the median/mean trial duration or Kaplan-Meier (KM) follow-up period. The 5-year absolute ASCVD rates by multiplying the annualized rate by 5, assuming consistent relative risk reductions during the trial as was observed in the moderate versus high intensity statin trials (**S2 Fig**) [12].

The previous analysis by Robinson and Stone grouped trial results by cardiovascular disease status and by the presence of diabetes, prediabetes/insulin resistance/metabolic syndrome, or the absence thereof [6]. In this updated systematic review, numerous other subgroups have been reported. In addition, the risk of the study populations in the trials of ezetimibe and PCSK9 mAbs trials were enhanced with additional risk characteristics, including recent acute coronary syndromes [13, 14]. Thus, due to the greater variety of high risk characteristics defining the subgroups and the higher risk patients entering the trials, the subgroups for this analysis were classified *a priori* into risk groups with annualized ASCVD rates of $\geq 4\%$, 3–3.9%, and <3% in the placebo/control group.

To summarize the log linear association for each of the 3 annualized ASCVD risk groups, least-squares meta-regression weighted by sample size for each subgroup was used to characterize log linear relationship between mean/median on-treatment LDL cholesterol and absolute ASCVD rates. All statistical analyses were performed using the Statistical Analysis System (SAS) statistical software package, version 9.4. SAS Institute Inc., Cary, NC, USA.

## Results

Characteristics of the main trials and their subgroups of moderate versus high intensity statin, ezetimibe versus placebo and PCSK9 mAbs versus placebo are provided in **Table 1**.

Abstraction results for the subgroups are included in the **S2 File**. In 2 trials of moderate versus high intensity statins, TNT and IDEAL [15, 16], 42 treatment subgroups were identified. For the single ezetimibe outcomes trial, IMPROVE-IT [14], 16 treatment subgroups were identified [17–19]. For the PCSK9 mAb trials [13], 38 treatment subgroups with ASCVD outcomes were identified from the FOURIER trial of evolocumab [20–26]. Because no ASCVD outcomes as defined above were reported for the PCSK9 mAb alirocumab trial, ODYSSEY OUTCOMES [27], a sensitivity analysis with the trial's primary endpoint of death, nonfatal MI, fatal or nonfatal stroke, or hospitalized unstable angina is reported in **S3 Fig**.

All participants had clinical ASCVD and >99% were receiving a statin, the majority at a high intensity. Subgroups were defined by baseline characteristics of age, diabetes status, type, timing, or number of clinical ASCVD events, statin intensity, level of LDL-C, high-sensitivity C-reactive protein, lipoprotein (a), or number of high risk characteristics. Subgroup analyses by gender were not identified. On-treatment LDL-C levels in the treatment and control groups ranged from 21 to 104 mg/dl, and were <100 mg/dl in all but 5 treatment subgroups.

The associations between on-treatment LDL-C and ASCVD event rates for the moderate versus high intensity, ezetimibe, and PCSK9 mAb trials plot as would be expected from the log linear associations between on-treatment LDL-C level and ASCVD event rate observed in the

**Table 1. Main trial and subgroup characteristics.**

| Trial | Trial population | Median/mean trial duration | Randomization groups | N | Mean/median on-treatment LDL-C | Subgroup-defining baseline characteristics |
|---|---|---|---|---|---|---|
| IDEAL | History of MI | 4.8 years | Simvastatin 20 mg | 4449 | 104 mg/dl | Age <65 years/Age 65–80 years<br>CKD/No CKD<br>PVD present/absent<br>Smoking status current, former, never |
| | | | Atorvastatin 80 mg | 4439 | 81 mg/dl | |
| TNT | Chronic CHD | 4.9 years | Atorvastatin 10 mg | 5006 | 101 mg/dl | Age <65/Age 65–75 years<br>CABG present/absent<br>CKD present/absent<br>DM present absent<br>DM+CKD<br>DM+ No CKD<br>Metabolic syndrome present/absent<br>Resistant HTN present/absent<br>Smoking status current, former, never |
| | | | Atorvastatin 80 mg | 4995 | 77 mg/dl | |
| IMPROVE-IT | Recent acute coronary syndrome on simvastatin | 6 years | Placebo | 9077 | 70 mg/dl | High risk (≥3 risk enhancers)*<br>Intermediate risk (2 risk enhancers)*<br>Low risk (0–1 risk enhancer)*<br>Diabetes present/absent<br>Age <65, 65–75, ≥75 years |
| | | | Ezetimibe 10 mg | 9067 | 54 mg/dl | |
| FOURIER | Chronic atherosclerotic cardiovascular disease on maximal statin | 2.2 years | Placebo | 13,780 | 92 mg/dl | Baseline LDL-C <70, ≥70 mg/dl<br>DM present/absent<br>hsCRP levels <1mg/dL, 1 to 3 mg/DL, > 3mg/dL<br>Lp(a) <37 nM, ≥37 nM<br>MI history present/absent<br>MI <2 years ago, ≥2 years ago<br>≥2 prior MI, 0–1 prior MI<br>Multi-vessel coronary artery disease present/absent<br>PVD present/absent<br>Statin therapy maximal/submaximal |
| | | | Evolocumab | 13,784 | 30 mg/dl | |
| ODYSSEY OUTCOMES | Acute coronary syndrome <1 year on maximal statin | 2.8 years | Placebo | 9462 | 96 mg/dl | Prior CABG, CABG at index event, no CABG<br>CHD only<br>CHD+PVD<br>CHD+CeVD<br>CHD+ PVD+ CeVD<br>DM, preDM, no DM<br>Genetic risk score low/high<br>2018 AHA/ACC guideline very high risk with multiple events, very high risk with single event, not very high risk |
| | | | Alirocumab | 9462 | 48 mg/dl | |

* Risk enhancers: Heart failure, hypertension, age ≥75 years, diabetes mellitus, prior stroke, prior coronary artery bypass grafting, peripheral arterial disease, estimated glomerular filtration rate <60 ml/min/1.73 m$^2$, current smoking

AHA/ACC American Heart Association/American College of Cardiology

CABG Coronary artery bypass grafting

CeVD Cerebrovascular disease

CHD Coronary heart disease

CKD Chronic kidney disease

DM Diabetes mellitus

FOURIER Further Cardiovascular Outcomes Research with PCSK9 Inhibition in Subjects with Elevated Risk

HTN Hypertension

IDEAL Incremental Decrease in End Points Through Aggressive Lipid Lowering

IMPROVE-IT Improved Reduction of Outcomes: Vytorin Efficacy International Trial

MI Myocardial infarction

PVD Peripheral vascular disease

TNT Treating to New Targets

statin versus placebo subgroup analysis [6] (**Fig 1**). Notably, the steepest risk reduction slopes are evident when LDL-C levels are >100 mg/dl, or in when annualized ASCVD risk is ≥4%. In contrast, in the high-risk groups, slope of the risk reduction is largely driven by the LDL-C level. In addition, the flattened slopes of the ezetimibe and PCSK9 mAB subgroups with <4% annualized ASCVD risk suggests much less relative or absolute risk reduction with ezetimibe or PCSK9 mAb added to background therapy in patient subgroups with an anticipated ASCVD risk <4% per year.

## Discussion

The evidence from statin, ezetimibe and PCSK9 mAb cardiovascular outcomes trials supports a log linear, or curvilinear, association between on-treatment LDL-C and ASCVD event rates. An attenuation of the association between LDL-C level and ASCVD event rates occurred with progressively lower LDL-C levels below 100 mg/dl unless annualized ASCVD risk was ≥4%. The subgroups with ≥4% annualized ASCVD risk are characterized by a large or active burden of ASCVD [such as polyvascular disease or myocardial infarction (MI) within 2 years] in the setting of a poorly controlled atherosclerotic milieu (multiple high risk characteristics, including diabetes, chronic kidney disease, suboptimal statin therapy, or age ≥75 years). These patients would be included in the "very high risk" (VHR) group of the 2018 AHA-ACC-Multi-Society Guidelines.(1) In the ODYSSEY OUTCOMES trial, those with recent acute coronary syndrome and dyslipidemia who were designated as very high risk had 2.7 times the rate of recurrent ischemic events and consistent with the findings in this analysis, derived a larger absolute risk reduction from treatment with alirocumab and a greater relative risk reduction when baseline LDL-C >100 mg/dl [28].

Previous analyses that have described a linear association between magnitude or percent LDL-C reduction relative risk reduction [5, 29]. This is due to the use of relative risk reductions, which plot linearly, but also to using a single effect estimates despite significant treatment heterogeneity between subgroups.

Duration of the PCSK9 mAb trials has been implicated as the reason for the apparent attenuation of benefit observed in the cardiovascular outcomes trials [5]. However, two PCSK9 mAb efficacy trials with baseline LDL-C levels of approximately 120 mg/dl of 11 and 18 months duration found 50% reductions in the relative risk of cardiovascular events [30, 31]. A sensitivity analysis using a 25% reduction in the relative risk of ASCVD each year over 5 years (reflecting the 25% ASCVD relative risk reduction observed in year 2 in the evolocumab trial [13], for a cumulative risk reduction of 25% over 5 years yielded very similar results (**S4 Fig**). A sensitivity analysis of subgroups from the alirocumab cardiovascular outcomes trial with a median 2.8 year follow-up [27] appeared similar, with modest slopes for the association between on-treatment LDL-C level and major adverse cardiovascular event rates observed even when annualized event rates exceeded 7% per year.

Why do our findings suggest diminishing returns from more aggressive LDL-C lowering, especially when for levels of LDL-C <100 mg/dl? All patients in the moderate versus high intensity statin trials, ezetimibe, and PCSK9 mAb trials had clinical ASCVD and received background statin therapy. While progressively lower achieved LDL-C <100 mg/dl is associated with progressively greater plaque regression, the magnitude of regression is still modest compared to the burden of plaque [32]. Patients with clinical SCVD remain at high risk of recurrent ASCVD events, often due to erosion of stable plaque in the setting of poorly controlled risk factors or large plaque burden [33, 34]. Pathophysiologic and imaging data, are supported by findings from the cardiovascular outcomes trials. The meta-analysis of the statin, ezetimibe, and PCSK9 mAb trials by mean baseline LDL-C level [7], and the ODYSSEY

OUTCOMES trial analysis by baseline LDL-C level [27]. Taken together, these data suggest that when LDL-C levels remain ≥100 mg/dl despite statin therapy, continued plaque progression and less plaque stabilization are more likely, progressively increasing the risk of an acute occlusive thrombus and fatal and nonfatal ASCVD events at higher LDL-C. Notaby, no reduction in cardiovascular or total mortality was found in the meta-analysis by baseline LDL-C level or ODYSSEY OUTCOMES trial when baseline LDL-C was <100 mg/dl. However, this analysis by subgroup does suggest that for patients at extremely high ASCVD risk ≥4%/year due to an extensive burden of ASCVD and poorly controlled risk factors, more aggressive LDL-C to below <100 mg/dl is more likely to provide a clinically meaningful risk reduction benefit than in patients at lower ASCVD risk.

Limitations of this analysis include reliance on trial rather than individual level data. Strengths include comparing multiple subgroups from trial populations with chronic ASCVD across a broad range of baseline LDL-C levels and time periods.

## Clinical implications

The log linear association between LDL-C and ASCVD risk reduction results in diminishing cardiovascular risk reduction benefits from intensifying LDL-C lowering below 100 mg/dl unless ASCVD risk is extremely high due to an extensive burden of atherosclerosis and poorly controlled risk factors. This has important implications for clinical practice, cost effectiveness, and clinical trial planning [35, 36]. The findings from this analysis support an evidence-based approach based on LDL-C and risk levels, which can be used to guide choice of the next therapy when considering multiple LDL-C lowering and other risk reduction therapies for a high risk patient.

In those with clinical ASCVD who have an LDL-C ≥100 mg/dl, further LDL-C reduction should be prioritized (**Fig 2**). These are the patients most likely to experience a meaningful clinical benefit from further LDL-C lowering that includes a reduction in the risk of all-cause and cardiovascular mortality as well as a greater relative risk reduction for a given magnitude of LDL-C lowering [7]. This translates into improved cost-effectiveness when LDL-C ≥100 mg/dl [35]. In a recent cost-effectiveness analysis of the ODYSSEY OUTCOMES trial of alirocumab versus in patients with an acute coronary syndrome in the previous year, at an acquisition price of US$5850, the incremental cost effectiveness ratio of alirocumab was US$41,000 per quality adjusted life-year (QALY) for those with LDL-C ≥100 mg/dl on maximal statin therapy, compared to US$ 299,400 per QALY when LDL-C was <100 mg/dl [36].

Moreover, the higher the absolute ASCVD risk, the greater the relative risk reduction from a given reduction in LDL-C, a relationship that is further amplified as the level of pre-treatment LDL-C increases. Therefeore, the next treatment prioirty in patient with LDL-C ≥100 mg/dl should therefore be increasing statin therapy to a high intensity statin as tolerated due to the extensive body of randomized trial evidence of greater coronary heart disease and stroke reductions with high compared to moderate intensity statin therapy [37]. In addition, heart failure hospitalizations were reduced by 26% in the high versus moderate intensity statin therapy group in patient with CHD [15].

In contrast, when LDL-C levels are <100 mg/dl on maximally tolerated statin therapy, mortality benefits are less likely to accrue from further LDL-C lowering, and there is an attenuation of the relative risk reduction in CHD events due to being in the flat part of the log linear curve [7]. As a result, cost-effectiveness worsens in this patient group compared to those with LDL-C ≥100 mg/dl [36]. In both large scale PCSK9 mAb trials involving very high risk secondary prevention patients, despite a large reduction in LDL-C to LDL-C levels of 30–50 mg/dl, the reduction in ASCVD endpoints was only 20% in the trial population overall [13, 27]. Based on

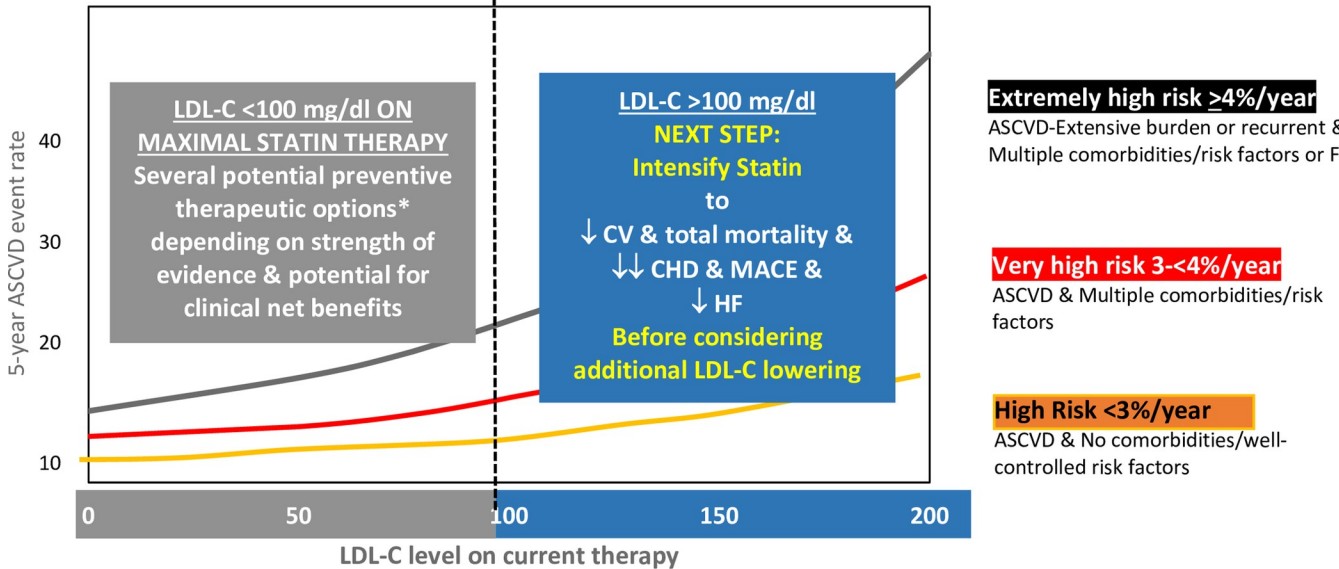

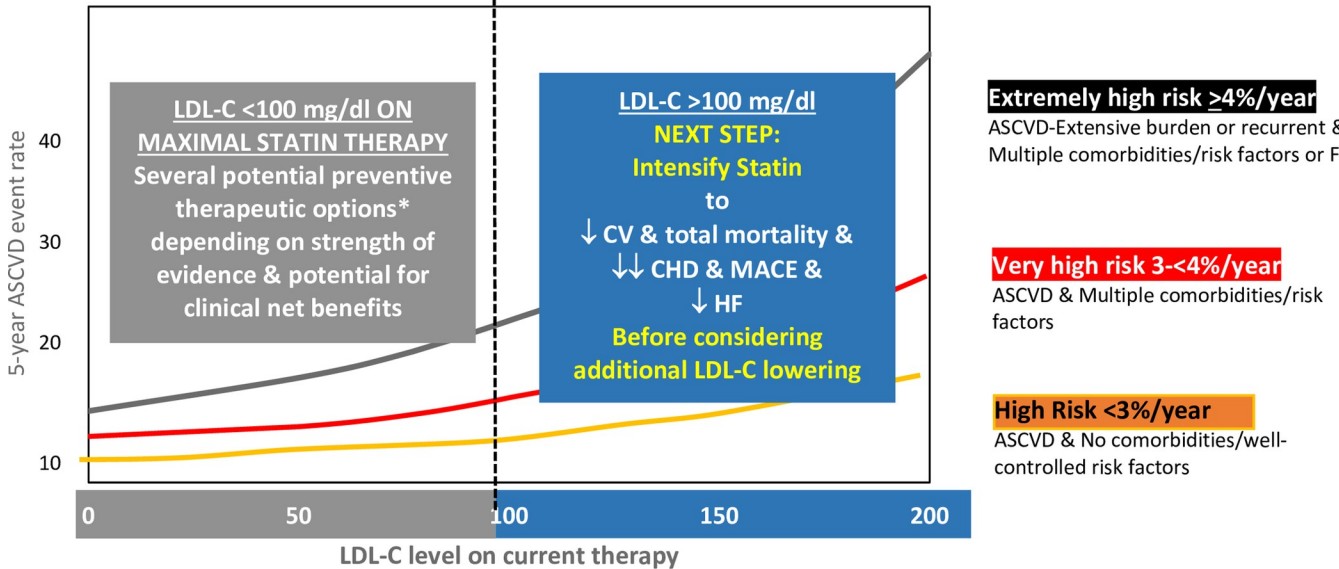

**Fig 2. Priorities for adding cardiovascular risk reduction therapies guided by LDL-C above or below 100 mg/dl on maximal statin and lifestyle therapy in patients with ASCVD.** When LDL-C levels are above 100 mg/dl, log linear association evidence for higher ASCVD risk groups and ASCVD risk reductions are substantial. When LDL-C is less than 100 mg, those patient groups with an extensive burden of atherosclerosis and multiple comorbidities, including familial hypercholesterolemia, are likely to have the greatest risk reduction from LDL-C lowering therapy.

randomized trial subgroup analyses, further LDL-C lowering with ezetimibe or a PCSK9 mAb appears to provide the a meaningful clinical benefit when baseline LDL-C levels are <100 mg/dl only in the extremely high risk ASCVD patients ≥4% risk of ASCVD per year, eg those with an extensive burden of atherosclerosis as well as multiple comorbidities or heterozygous familial hypercholesterolemia. This is the group of patients where PCSK9 mAbs are most likely to provide a reasonable value (<US $100,000/quality adjusted life year) [35].

In ASCVD patients with <4% risk of an ASCVD event per year and an LDL-C <100 mg/dl on maximally tolerated statin therapy, alternatives to further LDL-C lowering are reasonable to consider as a next step. Depending on patient characteristics, costs, potential for adverse effects, and patient preferences and priorities [38], options for the next secondary prevention therapy may include icosapent ethyl in patients with triglycerides ≥150 mg/dl [39], a sodium-

glucose cotransporter 2 (SGLT2) inhibitor or glucagon-like peptide 1 receptor (GLP-1) agonist in patients with diabetes or heart failure [40], or rivaroxaban in patients with coronary or peripheral artery disease at low bleeding risk [41, 42].

Notably, all of the recent cardiovascular outcomes trials were performed in the setting of very good risk factor control, and had mean baseline LDL-C levels <100 mg/dl. Thus, an evidence-based approach is to maximize statin therapy and reduce LDL-C to <100 mg/dl, where these new drugs have demonstrated additional efficacy. The benefit/risk profile of these new drugs may not be the same in patients with LDL-C levels >100 mg/dl [43].

In the REDUCE-IT trial, icosapent ethyl (a highly purified form of the omega-3 fatty acid eicosapentaenoic acid) 2 g twice was shown to reduce the risk of ASCVD events by 25%, and was recently approved by the Food and Drug Administration for ASCVD prevention in patients with ASCVD or diabetes with two risk factors who have triglyceride levels ≥150 mg/dl [39, 44]. In REDUCE-IT the elevated triglycerides likely served as a risk marker for a higher risk group, as benefit was irrespective of the baseline or attained triglyceride level.

Extensive data are rapidly accumulating that several sodium-glucose cotransporter 2 (SGLT2) inhibitors and glucagon-like peptide 1 receptor (GLP-1) agonists reduce ASCVD events in patients with diabetes [40, 45]. Some SGLT2 inhibitors have also been shown to reduce heart failure in patients with diabetes or heart failure, death, and renal outcomes and some GLP-1 agonists have been shown to reduce the risk of all-cause mortality and improve renal outcomes [40, 46–48].

In the COMPASS trial, rivaroxaban 2.5 mg twice daily added to aspirin reduced the risk of major cardiovascular events by 25% and all-cause mortality by 18% compared to aspirin alone in patients with coronary heart disease and at least one high risk comorbidity or peripheral arterial disease treated on average for 23 months [49]. The greatest net benefit in ASCVD risk reduction was observed in those with multiple high risk characteristics [42]. Risk prediction models have been developed to predict patients likely to experience the most benefit and the lowest risk of bleeding [50].

## Pharmacoeconomic implications

The log linear association between baseline LDL-C level and ASCVD risk reduction suggests that at any given level of ASCVD risk, additive LDL-C lowering drug therapy is likely to prove less cost-effective in those with lower LDL-C levels [35]. However, analyses of the statin, PCSK9 mAb, and ezetimibe cardiovascular outcomes trials reveal subgroups of patients at very high or extremely high ASCVD risk with lower LDL-C levels for whom further LDL-C lowering may have similar cost-effectiveness as in lower risk patients with higher LDL-C levels [35]. Because the time horizons for estimating cost-effectiveness are typically similar to the 2 to 5-year follow-up periods of the clinical trials, estimates of cost-effectiveness over 10 years or a lifetime are likely different. For example, initiation of PCSK9 mAb therapy at younger ages in those with higher risk factor burden and LDL-C levels may provide the greatest lifetime benefit [51]. Acquisition pricing of forthcoming LDL-C lowering drugs will strongly influence the cost-effectiveness of those treatments for various patient risk groups at various baseline LDL-C levels. It should also be noted that estimates of cost-effectiveness for other cardiovascular prevention drugs should be based on the absolute risk of patient groups who have LDL-C levels close to the level that was achieved in the cardiovascular outcomes, typically 60–70 mg/dl.

## Conclusions

The results from cardiovascular outcomes trials of statins, ezetimibe and PCSK9 mAbs support a log linear association between on-treatment LDL-C and ASCVD event rates. This translates

into diminishing returns from very aggressively reducing LDL-C levels. When LDL-C levels are >100 mg/dl, increasingly greater reductions in cardiovascular and all-cause mortality as well as CHD events occur. Thus, ASCVD patients with LDL-C >100 mg/dl have the most relative risk reduction benefit from further LDL-C lowering, and statin intensification and additional LDL-C lowering therapy if needed should be next step in risk reduction therapy. In addition, ASCVD patients at the highest ASCVD risk (≥4%/year ASCVD risk) experience less flattening of the log linear curve when LDL-C levels are <100 mg/dl, and so may still experience clinically significant benefits from further lowering LDL-C. For ASCVD patients with <4%/year ASCVD risk and LDL-C <100 mg/dl, other secondary prevention options as the next step may provide greater ASCVD, mortality, and non-cardiovascular benefits.

Future research is needed to determine how best to optimize treatment regimens based on an individual patient's characteristics, preferences and cost effectiveness.

## Supporting information

**S1 Fig. Rate of incident MACE per 100 patient years by average achieved on-treatment LDL-D levels in patients with ASCVD with and without comorbidities in a pooled analysis of alirocumab Phase 3a trials.**
(RTF)

**S2 Fig. Reductions in the risk of major cardiovascular events per 39 mg/dl (1 mmol) reduction in LDL-C in trials of moderate versus high intensity statin therapy.**
(RTF)

**S3 Fig. Sensitivity analysis of the MACE primary outcome for subgroups from the ODYSSEY OUTCOMES trial.**
(RTF)

**S4 Fig. Sensitivity analysis using a 25% relative risk reduction for ASCVD in subgroups of the FOURIER trial.**
(RTF)

**S1 Table. PRISMA checklist.**
(PDF)

**S1 File. Methods for updated systematic review.**
(PDF)

**S2 File. Systematic review—results.**
(PDF)

## Author Contributions

**Conceptualization:** Jennifer G. Robinson, C. Noel Bairey Merz.

**Data curation:** Jennifer G. Robinson, Manju Bengaluru Jayanna.

**Formal analysis:** Jennifer G. Robinson, Manju Bengaluru Jayanna.

**Methodology:** Jennifer G. Robinson.

**Project administration:** Jennifer G. Robinson.

**Resources:** Jennifer G. Robinson.

**Software:** Manju Bengaluru Jayanna.

**Supervision:** Jennifer G. Robinson.

**Validation:** Jennifer G. Robinson, Manju Bengaluru Jayanna.

**Visualization:** Jennifer G. Robinson.

**Writing – original draft:** Jennifer G. Robinson.

**Writing – review & editing:** Jennifer G. Robinson, Manju Bengaluru Jayanna, C. Noel Bairey Merz, Neil J. Stone.

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
