## [Decision Letter · Decision Letter 0]

16 Sep 2020

PONE-D-20-13528

Clinical implications of the Log Linear Association Between LDL-C Lowering and Cardiovascular Risk Reduction: Greatest Benefits When LDL-C >100 mg/dl

PLOS ONE

Dear Dr. Robinson,

Thank you for submitting your manuscript to PLOS ONE. After careful consideration, we feel that it has merit but does not fully meet PLOS ONE’s publication criteria as it currently stands. Therefore, we invite you to submit a revised version of the manuscript that addresses the points raised during the review process.

We find you manuscript interesting, but the reviewers have raised some issues concerning statistics. Check them carefully and reply clearly and mark the changes.

We look forward to receiving your revised manuscript.

Kind regards,

Katriina Aalto-Setala, Professor

Academic Editor

PLOS ONE

Journal Requirements:

2. Thank you for including your competig interests statement;"I have read the journal's policy and the authors of this manuscript have the following competing interests:

Jennifer G Robinson MD MPH has received research grants to Institution: Acasti, Amarin, Amgen, Astra-Zeneca, Esperion, The Medicines Company, Merck, Novartis, Novo-Nordisk, Regeneron, Sanofi, Takeda and served as a consultant for The Medicines Company, Novartis, and Pfizer.

Manju Bengularu Jayanna MBBS None.

C Noel Bairey Merz MD has received research grants to Institution: Caladrius, Gilead, Sanofi ACT, and served as a consultant for iRhythm, Sanofi Vascular, and Abbott Diagnostics.

Neil J Stone MD None."

Reviewers' comments:

Reviewer's Responses to Questions

**Comments to the Author**

1. Is the manuscript technically sound, and do the data support the conclusions?

Reviewer #1: Yes

Reviewer #2: Yes

2. Has the statistical analysis been performed appropriately and rigorously? 

Reviewer #1: Yes

Reviewer #2: Yes

3. Have the authors made all data underlying the findings in their manuscript fully available?

Reviewer #1: Yes

Reviewer #2: Yes

4. Is the manuscript presented in an intelligible fashion and written in standard English?

Reviewer #1: Yes

Reviewer #2: Yes

5. Review Comments to the Author

Reviewer #1: I’ve read with attention the paper of Robinson et al. that is potentially of interest. The background and aim of the study have been clearly defined. The methodology applied is overall correct, the results are reliable and adequately discussed. The authors should only add a couple of lines of pharmacoeconomic aspects related to their findings. Moreover, they should stress the concept that this observation could loose a bit of its strenght if considering long-term treatments beyond the trial lenght (I mean, decades)

Reviewer #2: Interesting paper

methods. Which is the difference betwee cv and chd death?

methods: the authors spoke about weighted analysis but this should be added also in figure

methods: kind of analysis is not clear. Please be more specific. Do authors performed a meta-regression?

results: from figure 1, it appears that the risk for "high risk" patients is more relevant for LDL more than 70 rather thano 100. Please clarify

6. PLOS authors have the option to publish the peer review history of their article (what does this mean?). If published, this will include your full peer review and any attached files.

Reviewer #1: No

Reviewer #2: **Yes: **Fabrizio D'Ascenzo

---

## [Author Response · Author response to Decision Letter 0]

18 Sep 2020

Response to reviewers

We thank the reviewers for their positive review of our paper, and the additional insights that they provided into our analyses.

Reviewer #1: I’ve read with attention the paper of Robinson et al. that is potentially of interest. The background and aim of the study have been clearly defined. The methodology applied is overall correct, the results are reliable and adequately discussed. The authors should only add a couple of lines of pharmacoeconomic aspects related to their findings. Moreover, they should stress the concept that this observation could loose a bit of its strenght if considering long-term treatments beyond the trial lenght (I mean, decades)

Response: We thank reviewer #1 for the recommending further comment on the pharmacoeconomic implications of our analysis. We agree this is an important additional perspective and took the opportunity to highlight it as such by adding this call-out paragraph and adding reference 51:

PHARMACOECONOMIC IMPLICATIONS

The log linear association between baseline LDL-C level and ASCVD risk reduction suggests that at any given level of ASCVD risk, additive LDL-C lowering drug therapy is likely to prove less cost-effective in those with lower LDL-C levels.35 However, analyses of the statin, PCSK9 mAb, and ezetimibe cardiovascular outcomes trials reveal subgroups of patients at very high or extremely high ASCVD risk with lower LDL-C levels for whom further LDL-C lowering may have similar cost-effectiveness as in lower risk patients with higher LDL-C levels.35 Because the time horizons for estimating cost-effectiveness are typically similar to the 2 to 5-year follow-up periods of the clinical trials, estimates of cost-effectiveness over 10 years or a lifetime are likely different. For example, initiation of PCSK9 mAb therapy at younger ages in those with higher risk factor burden and LDL-C levels may provide the greatest lifetime benefit.51 Acquisition pricing of forthcoming LDL-C lowering drugs will strongly influence the cost-effectiveness of those treatments for various patient risk groups at various baseline LDL-C levels. It should also be noted that estimates of cost-effectiveness for other cardiovascular prevention drugs should be based on the absolute risk of patient groups who have LDL-C levels close to the level that was achieved in the cardiovascular outcomes, typically 60-70 mg/dl. 

51. Kaasenbrod L, Ray KK, Boekhoeldt M, Smulders YO, LaRosa JC, Kastelein JJP, van der Graaf Y, Dorresteijn JAN, Visseren FL. Estimated individual lifetime benefit from PCSK9 inhibition in statin-treated patients with coronary artery disease. Heart 2018; 104:1699-1705.

Reviewer #2: Interesting paper

Response: Thank you

methods. Which is the difference betwee cv and chd death?

Response: Cardiovascular include coronary and cerebrovascular deaths, whereas CHD death includes only those of cardiac origin such as due to myocardial infarction.

This clarification was added:

Cardiovascular death includes deaths from CHD and stroke. 

methods: the authors spoke about weighted analysis but this should be added also in figure

Response: Thank you for bringing this to our attention. 

Added to Figure legend: Line represents log linear regression line for relationship between on-treatment LDL-C level and cardiovascular event rate weighted by group size. 

methods: kind of analysis is not clear. Please be more specific. Do authors performed a meta-regression?

Response: Thank you for the opportunity to clarify. Yes, this was a meta-regression. 

Revised to read: To summarize the log linear association for each of the 3 annualized ASCVD risk groups, least-squares meta-regression weighted by sample size for each subgroup

results: from figure 1, it appears that the risk for "high risk" patients is more relevant for LDL more than 70 rather thano 100. Please clarify

Response: Thank you for the opportunity to add this additional important insight to the Results section.

Revised to read: Notably, the steepest risk reduction slopes are evident when LDL-C levels are >100 mg/dl, or in when annualized ASCVD risk is >4%. In contrast, in the high-risk groups, slope of the risk reduction is largely driven by the LDL-C level. 

---

## [Editor Report · Decision Letter 1]

22 Sep 2020

Clinical implications of the Log Linear Association Between LDL-C Lowering and Cardiovascular Risk Reduction: Greatest Benefits When LDL-C >100 mg/dl

PONE-D-20-13528R1

Dear Dr. Robinson,

We’re pleased to inform you that your manuscript has been judged scientifically suitable for publication and will be formally accepted for publication once it meets all outstanding technical requirements.

Kind regards,

Katriina Aalto-Setala, Professor

Academic Editor

PLOS ONE
---

## [Editor Report · Acceptance letter]

6 Oct 2020

PONE-D-20-13528R1 

Clinical implications of the Log Linear Association Between LDL-C Lowering and Cardiovascular Risk Reduction: Greatest Benefits When LDL-C >100 mg/dl  

Dear Dr. Robinson:

I'm pleased to inform you that your manuscript has been deemed suitable for publication in PLOS ONE. Congratulations! Your manuscript is now with our production department. 

Kind regards, 

on behalf of

Dr Katriina Aalto-Setala 

Academic Editor

PLOS ONE